# Spatial analysis of malaria hotspots in Dilla sub-watershed: Western Ethiopia

**Gemechu Y. Ofgeha** *

Department of Geography and Environmental Studies, Wallagga University, Gimbi, Ethiopia

* dandigemechu11@gmail.com

**Data Availability Statement:** All relevant data are within the paper and its Supporting Information file.

**Funding:** This work was supported by Wollega University, Ethiopia. The funder has no role in the

## Abstract

### Background

This study aimed to examine the spatial variations in malaria hotspots along Dilla sub-watershed in western Ethiopia based on environmental factors for the prevalence; and compare the risk level along with districts and their respective kebele. The purpose was to identify the extent of the community's exposure to the risk of malaria due to their geographical and biophysical situations, and the results contribute to proactive interventions to halt the impacts.

### Methods

The descriptive survey design was used in this study. Ethiopia Central Statistical Agency based meteorological data, digital elevation model, and soil and hydrological data were integrated with other primary data such as the observations of the study area for ground truthing. The spatial analysis tools and software were used for watershed delineation, generating malaria risk map for all variables, reclassification of factors, weighted overlay analysis, and generation of risk maps.

### Results

The findings of the study reveal that the significant spatial variations in magnitudes of malaria risk have persisted in the watershed due to discrepancy in their geographical and biophysical situations. Accordingly, significant areas in most of the districts in the watershed are characterized by high and moderate in malaria risks. In general, out of the total area of the watershed which accounts 2773 km$^2$, about 54.8% (1522km$^2$) identified as high and moderate malaria risk area. These areas are explicitly identified and mapped along with the districts and kebele in the watershed to make the result suitable for planning proactive interventions and other decision making.

### Conclusions

The research output may help the government and humanitarian organizations to prioritize the interventions based on identified spatial situations in severity of malaria risks. The study was aimed only for hotspot analysis which may not provide inclusive account for community's vulnerability to malaria. Thus, the findings in this study needs to be integrated with the

study design, data collection and analysis, decision
to publish, or preparation of the manuscript.

**Competing interests:** The authors have declared
that no competing interests exist.

socio-economic and other relevant data for better malaria management in the area. There-
fore, future research should comprehend the analysis of vulnerability to the impacts of
malaria through integrating the level of exposure to the risk, for instance identified in this
study, with factors of sensitivity and adaptation capacity of the local community.

## Introduction

Malaria is one of the world's most serious and complex public health problems, and it remains
the cause of a substantial number of deaths in Ethiopia [1]. Malaria occurs in over 100 coun-
tries and more than 40% of the people in the world are at risk. The World Health Organization
estimates that 300–500 million cases of malaria occur worldwide each year, resulting in the
death of over two million people. Malaria is a leading cause of morbidity and mortality in the
developing world, especially in Africa where the transmission rates are highest and where it is
considered to be a major impediment to economic development. The majority of malaria
deaths occur in sub-Saharan Africa, where malaria also presents major obstacles to social and
economic development. It has been estimated to cost Africa more than US$ 12 billion every
year in lost Gross Domestic Product (GDP), even though it could be controlled for a fraction
of that sum [2, 3].

The malaria epidemic and its far-reaching socio-economic impacts are significant, and
cases are increasing at an alarming rate in sub-Saharan African countries [4]. However,
malaria risk varies significantly from place to place and season to season due to weather and
climate controls, primarily rainfall and temperature, which are further influenced by, among
other things, altitude in Ethiopia [5]. Therefore, biophysical factors such as topography, cli-
mate, soil and drainage patterns made the ground fertile for the reproduction of the epidemic.
The estimated three-fourth of the land and more than 68% of the population in Ethiopia live
below 2000 meters, with consequent variation in minimum and maximum temperatures,
which is particularly malarious in the country. These make malaria one of the health problems
seeking attention in the country [6].

Several studies revealed that the lack of implementing a systematic approach to examining
the hotspots and preventive plans because of the low socio-economic status of the society
aggravated the impacts mainly among the poor [4, 7, 8], while the situations in many parts of
Ethiopia and the study area is not exceptional. As a result, the cost of preventive plans and
medical treatment becomes an enormous burden on the GDP [9]. At the same time, the eco-
nomic productivity of the individual has declined due to malaria infection. It has been esti-
mated that the economic burden due to malaria accounts for a 1.3% reduction in the annual
economic growth rate of Ethiopia. These conditions seek scientific investigations to provide
appropriate information for decision makings. Particularly, the application of spatial technolo-
gies has paramount importance to provide means for assessment, monitoring, and modeling
of geographical and biophysical variables over a large area and through time to analyze malaria
risks [10].

Spatio-temporal variability in bio-physical factors significantly determines malaria distribu-
tion in Ethiopia, while prevention and remedial measures have been taking place by the gov-
ernment and other stakeholders without due concern for these factors. However, the analysis
of the spatial variability of these factors and how the conditions contributed to the distribution
of the disease seek due attention. Specifically, Wollega University's Department of Social
Work, in collaboration with Gimbi hospital, attempted to provide community service to

enhance community engagement in controlling the epidemics in the West Wallagga zone, focusing on HIV/AIDS and malaria in the last five years. During the implementation of the project, the researcher, as the delegated person to facilitate the project, observed some of the challenges faced by the experts, mainly in screening the hot spots of malaria to prioritize the areas of intervention. The experts were forced to use the hospitals' and health centers' case reports in selecting the areas of implementation. However, vulnerability to malaria doesn't imply the level of the disease prevalence/the risk level, and difficult to genuinely selecting of the areas to implement the project. Therefore, identifying the community's level of exposure to malaria due to their geographical and biophysical positions has paramount importance for such preventive interventions.

Malaria breeding and transmission are associated with physical factors such as altitude, temperature, and rainfall, accessibility to rivers, lakes, and swamps [6] which could have been studied by GIS applications but still not widely applied in the country. A detailed understanding of what drives heterogeneities in the distribution of mosquitoes and mosquito-borne diseases could be analyzed by this tool which can help us to design better, more efficient control programs that maximize the use of limited resources [11].

Spatial technology has clearly defined the epidemiology of disease Vis environmental factors by identifying the spatial limits of the disease prevalence and risk mapping with relevant risk factors [2, 10]. Specifically, Geographic Information Systems (GIS) are powerful computer mapping and analysis systems for studying spatial patterns and processes and apply to numerous disciplines including the study of mosquito ecology. The distribution of mosquitoes could be studied by this technology depending on the spatial distribution of their larval breeding sites, their flight range, and their preferred hosts. These are all heterogeneous in space and time, and GIS, therefore, has many potential applications to the study of mosquitoes and the diseases they transmit. Therefore, GIS application has suitable tools to map and analyze the spatial distribution of mosquitoes and to assess the ecological factors that contribute to observed distributions [2, 12, 13]. However, the potential application of GIS to epidemiological studies has been shown by only recent studies and very few applications have been made, even though epidemiological data have spatial components [14]. Nevertheless, GIS and remote sensing could be fantastic tool for disease management as it has an inherent ability to manage spatial, non-spatial, and temporal data to investigate associations between environmental variables and the distribution of the species responsible for malaria transmission [2, 15], the application is not yet fully utilized by the stakeholders.

Moreover, the study watershed was selected with the strong assumption that the area is one of the malaria hotspots in the country due to facilitating physical and socio-economic conditions. The previous studies revealed that the distribution and transmission of malaria in Ethiopia varies from place to place largely determined by altitude through its effect on temperature and rainfall. Thus, the lowland and midlands of western Ethiopia, where the study area is located are among the vulnerable region in the country. This is the ground for the purposive selection of the watershed for this study. The watershed encompasses eight districts of the west Wallagga zone and four districts of Qellem Wallagga. The physical setting of the study area highly facilitates the malaria parasite breeding with a dominant altitude of less than 2000 meters (ranges from 1259–2499), rainfalls range from 1292 to 1738mm, and average annual temperature ranges from 19-200c. Therefore, identifying and mapping malaria-prone areas, and characterizing the areas according to their risk level in the watershed is vital in prioritizing areas of interventions for health facilities, controlling and monitoring the disease. Likewise, the identification of the level of incidents could enhance the effectiveness of prevention efforts and will substantially reduce costs of prevention by efficiently targeting high and moderate-risk areas.

The objective of the study is to generate malaria risk map of Dilla watershed based on environmental factors responsible for malaria for malaria prevalence and compare malaria risk levels among the districts and their respective kebele is the lowest administrative tier at grassroots level in Ethiopia; it may be equated with parish/county to recommend the priority areas for prevention, controlling, and monitoring activities by concerned bodies.

The audiences for this study will be from both academic and non-academic areas which will benefit all of these audiences for various purposes. The research will fill the gap in identifying malaria risky areas and vulnerability analysis which traditionally take place based on health centers' reports. Moreover, this research work will contribute to identifying malaria hot spots for the community service project intended by the University in the last five years because the Dilla watershed for which this research work is proposed covers about 12 districts of West and Qellem Wallagga zones; covering the largest part of Wallagga University's research and community service geographical mandates.

## Conceptual framework of the study

The hotspot of malaria is governed by a large number of factors relating to the parasite, the vector and the socio-economic condition of the communities [16]. The biophysical factors affecting habitat and breeding sites of the *Anopheles* mosquito vectors such as temperature, precipitation, humidity, presence of stagnant water pools, vegetation are related with other factors such as elevation and slope of the area. Specifically, temperature and rainfall act as limiting factors on the development of *Anopheles* mosquitoes, which are the intermediate hosts in the transmission of malaria parasites. About 70–90% malaria risk is associated with environmental factors, which in turn influence the abundance and survival of the vectors [17].

In this study, the physical factors identified in previous research for their significant contributions for the prevalence of malaria distribution were identified with great care, and by critical analysis of studies conducted in Ethiopia. The standard to determine and rate the effect of each factor is also based on these studies [5, 6, 18]. The factors are described below:

### Temperature

Temperature is one of the key environmental contributors to mosquito propagation [19]. High temperature speeds up the development of the life cycle of a mosquito and accelerates the length of the development of the life cycle of malaria parasite within the mosquito host. The optimum temperature for development of malaria parasite is between 25°C to 30°C. At lower temperature less than 16°C, the larval and pupal stages of mosquitoes take longer time to complete sporogony cycle and below 16°C the sporogony ceases. However, with an increase in temperature sporogonic period become shortened and the Plasmodium parasite within the vector increase and effective up to about 30°C. Contrary to this, increased temperature above 30°C has negative impact on the survival of the vector [8, 20].

In general, the temperature less than 16°C and above 30°C, are less conducive for malaria breeding. Therefore, based on these literatures the temperature based malaria hot spot of the study area classified as high and very high for 19–21°C and 21–24°C respectively; believing that the temperature increasing more than 25°C will gradually decreases the breeding (The detail of the standard used is described in the Table 1).

### Rainfall

The association between rainfall and malaria epidemics has been recognized in various literatures [20, 21]. But it can be seen in two ways. First, increasing precipitation may increase vector populations in many circumstances by increasing available anopheles breeding sites.

**Table 1. Standardized value assigned for physical factors of malaria risk level.**

| S.N | Factors | Unit | Risk Level | | | | |
|-----|---------|------|------|------|------|------|------|
|     |         |      | R1 | R2 | R3 | R4 | R5 |
| 1 | Temperature | ˚C | | | | 19–21 | 21–24 |
| 2 | Rainfall | Mm | | | | 1504–1600 | >1600 |
| 3 | Elevation | M | | | > 2000 | 1500–2000 | 1317–1500 |
| 4 | Slope | % | >30 | 15–30 | 8–15 | 5–8 | 0–5 |
| 5 | Soil drainage | Water Drainage | | | Moderate | Poor | Very poor |
| 6 | Proximity to River | Km | | | 2–5 | 1–2 | < 1 |

R1, R2, R3, R4 and R5 represent very low, low, moderate, high and very high level, respectively

Sources: Modified from [18, 20, 29].

Excessive rainfall in warm, arid areas can lead to increased transmission due to creation of vector breeding sites [22]. Apart from creating mosquito breeding sites, rainfall also affects malaria transmission through increasing humidity, which in turn will help to increase the longevity of the adult vectors Second, excessive rains may also have the opposite effect by flushing out small breeding sites, such as ditches or pools or by decreasing the temperature, which in regions of higher altitude can stop malaria transmission [23].

Based on this information, it is possible to identify the significant amount of rainfall support malaria breeding while other physical factors are highly associated with its effectiveness. Areas with annual rainfall amount greater than 1000 mm are malarious and have intense malaria transmission, but areas with rainfall amount between 500 and 1000 mm have seasonal transmission [22, 23]. Moreover, very high rainfall is not suitable for vector immature stages, areas having >1600 mm annual rainfall is unfavorable for mosquito breeding. In general, rainfall based mosquito breeding zones are classified as high and very high for rainfall amount of 1504–1600 mm and >1600 mm respectively.

## Elevation

Malariologists working in the field in the first half of this century, in the decades following the elucidation of the malaria cycle in man and mosquitoes, appreciated that it was a focal disease and that the topography of the land was an important consideration in understanding the local epidemiological situation [2]. Topography (particularly altitude and slope) are identified by [13, 17] as an important factor in understanding the malaria epidemiological situation at local scale.

In Ethiopia, it is locally well-known that the prevalence of malaria parasites in people varies with altitude. People at low lands have significantly higher prevalence of malaria than those in middle and highlands. Therefore, altitude is significant in determining the distribution of malaria and its seasonal impact on many parts of the World. Based on altitude, traditionally Ethiopia is divided in five agro-ecological climatic zones. These are cold moist, cool humid, cool sub-humid, warm semi-arid, and hot arid locally known as *Baddaa Dilallaa*, *Baddaa*, *Badda Daree*, *Gammojjii and Gammoojjii Ho'aa* respectively in the study area. In Ethiopia in general and in the study area in particular, both *Baddaa Dilallaa and Baddaa* zone with elevation above 2500 meter above mean sea level is malaria free even though the dynamics is high due to climate change and variability. However, malaria frequently occurs in areas below 2000 meters elevation and the transmission is very intense in areas below 1500 meters elevation [5, 24]. Particularly, in some parts of *Badda Daree*, and in most parts of *Gammojjii*. Contrary to this, the areas of *Gammoojjii Ho'aa* have an altitude < 500 meter above mean sea level (masl)

and annual rainfall of < 900mm is not suitable for mosquito breeding as a result of low annual rainfall amount and very high temperature above 30˚C.Therefore, the study area is classified in to three classes as 1317–1500 masl, 1500–2000 m and > 2000 meter for very high, high and low elevation based malaria risks respectively.

## Slope

Malaria breeding is affected by slope of the land [15, 24]. Mosquito larvae need stagnant water pools to survive, and these pools are less likely to form in steep slope areas. Moreover, larvae developing in water pools in sloped areas are more likely to be washed away during downpours [25]. Thus, sloped areas make poor mosquito breeding ground, reducing the threat of malaria transmission because of low water stagnation. In general, the steeply the slope, the low mosquito prevalence and vice versa are well-known. Therefore, the slope classes 0–5%, 5–8%, 8–15%, 15–30% and >30% are assigned for very high, high, moderate, low and very low slope based risk of malaria respectively.

## Proximity to water bodies

Surface water provides the habitat for the juvenile stages (egg, larvae, and pupae) of malaria vectors. The state of small water bodies and wetlands is very useful source of malaria vectors [25, 26]. Seasonal variation in volume of water in Ethiopia results the variation in malaria breeding situation spatially and temporally. The rainy season for both the rivers and lakes lead to flooding which hinders vectors of malaria. However, during dry season their volume is decreased and they create different pockets of water body that is favorable ground for the breeding of mosquito. Therefore, even though the effect is varying seasonally, accessibility to water bodies aggravate the prevalence of malaria. The water body selected for this project could be river, lake and swamps. After careful analysis of previous studies conducted in Ethiopia and other parts of the world [20, 24, 27], the value is assigned for these variables as follows: Areas found < 1 km, 1–2 km and above 2–5 km from rivers and canals assigned as very high, high and low malaria risk level respectively.

## Soil moisture holding capacity

Soil moisture holding capacity/permeability is important factor determine mosquito breeding [28]. Poorly drained soils is believed to facilitate water stagnation and create conducive conditions for mosquito breeding and thus, favorable for malaria outbreak. Well drained soil doesn't allow water stagnation, so it creates unfavorable condition for anopheles breeding. Soil drainage classification of Ethiopia made by CSA, were used in this study. Based on review of relevant research findings [20, 23], the areas of very poor, poor, moderate, and high soil moisture holding capacity are assigned as low, high and very high malaria risk areas respectively.

## Research methods and materials

### Description of the study watershed

Dilla is a sub-watershed of Dabbus, which is one of the significant watersheds of the Blue Nile in the South Western and Western parts of the country. It is the largest river sub-basin whose watershed covers 12 districts out of 22 in the Dabbus Sub-basin of the Blue Nile Basin (Fig 1). Astronomically, the watershed lies between 9˚03'N -9˚11'N and 34˚36' - 35055'E which consists of eight districts of the West Wallaga zone, namely Babo Gambel, Jarso, Aira, Guliso, Yubdo, Boji Chokorsa, Boji Dirmaji, and Najo districts, and four districts of the Qellem Wallaga Zone, namely Gawo Kebe, Dale Wabera, Dale Sadi, and Lalo Kile.

## Map of the study area

**Fig 1. Map of the study area.** Source: Ethiopian Central Statistical Agency.

The watershed has total areas of 2773 Km². The altitude in this watershed ranges approximately between 1317 masl and 2405 masl. The highlands in the north-eastern part of the sub basin are higher in altitude, greater than 1800masl whereas the western part has relatively lower altitude up to 1317 masl. The rainfall distribution is associated with altitude that the areas of high altitude in north-eastern and eastern part of the watershed experiences relatively highest rainfall up to 1723mm whereas temperature distribution is generalized as 19˚c based on available data of CSA, 2007.

Accordingly, the watershed is characterized by hot to warm moist and sub humid lowlands and, the eastern and southern highlands the being tepid to cool sub humid mid highlands [18]. The most dominant soil in the watershed is Alisols and Nitosols, with the occurrence of Vertisols, Fluvisiols, and Acrisols.

### Research design

Descriptive survey design was used in this study. Central Statistical Agency based data were obtained to identify factors determine malaria distributions in the watershed. Among others, the pertinent environmental factors determining mosquito breeding and malaria transmission were identified for the study [3, 8, 18]. Accordingly, elevation, temperature, slope, soil drainage, rainfall and proximity to water bodies were selected. Malaria risk map were produced for all variables separately. Finally, all maps were overlaid to produce malaria risk area and used to compare the risk level of districts in general and their respective kebeles in particular in the study watershed. Field observations were used to understand the biophysical situations of the study area in this context, and the data were used for ground truthing.

## Sources and types of data

The CSA based meteorological data, digital elevation model, soil data and hydrological data were used to derive rainfall, temperature, elevation, slope, soil drainage and proximity to water body of the watershed to produce temperature based malaria risk map, rainfall based malaria risk map, altitude based malaria risk map, slope based malaria risk map, soil drainage based malaria risk map and proximity to water body based malaria risk map. The field observation data was used for ground truthing, and substantiate biophysical data.

## Description of variables and assigned value

As described in section 2, six biophysical factors (independent variables) are identified for analyzing and mapping of malaria hot spots (dependent variable) in the study watershed. Thus, Malaria Risk = function (the impact of rainfall, temperature, altitude, slope, soil drainage and proximity to river based malaria risks). The value of each variable was assigned according to predetermined standard derived from the studies conducted in Ethiopia with great caution and experiences of the study area (Table 1).

## Methods of data analysis

Arc SWAT for watershed delineation, and Arc GIS 10.3 was used for generating malaria risk map for all variables, reclassification of factors, for weighted overlay analysis and generation of risk maps through the application of spatial analysis tools. In this study, the analytical hierarchical process (AHP) was applied. It is a multi-criteria decision method that represents an issue using hierarchical structures and makes judgments based on expert input to determine priority scales. The mapping weight or relevance of each individual malaria risk factor was determined using AHP in this study [21]. To determine the weights of each factor, pairwise comparison matrix was created for each of the input parameters, and then the relative weights of each input parameter were determined.

Weighting of the impact of each variable on malaria distribution is one of the important issues in malaria hotspot analysis because all variables may not affect the distribution equally. In this regards, similar to most research conducted in Ethiopia, the hospitals report were used to substantiate the decision on the weight for the factors (Table 2). So the rating or assigning weights for the factors depends on previous research, and knowledge of the study area [11, 20, 27].

Malaria sensitivity of the study area was classified for all variables and presented in the map separately based on the value assigned in the above table (Table 1). The map of each variable which was classified according to predetermined value was reclassified by a spatial analysis tool to facilitate weighted overlie analysis. Because each variable has its unit of measurement which should be converted to an integer value (1 to 5) enables us to the overlying activity.

**Table 2. Weight assigned for the factors.**

| S.N | Factors | Assigned Weight /100% |
|---|---|---|
| 1 | Temperature | 25 |
| 2 | Rainfall | 20 |
| 3 | Elevation | 22 |
| 4 | Slope | 11 |
| 5 | Soil drainage | 10 |
| 6 | Proximity to River | 12 |

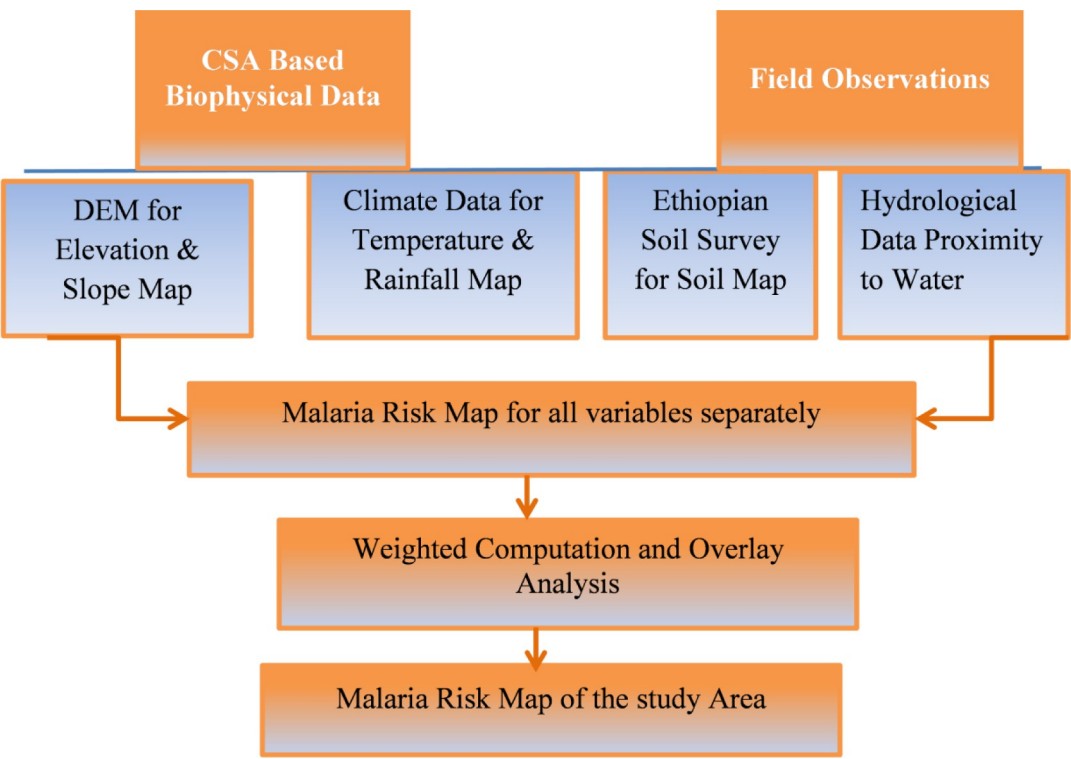

**Fig 2. Analysis flow chart.**

Finally, the map displaying the integrated effects of all variables was produced based on the weight assigned for the factors out of 100% (Table 2).

Finally, the malaria risk map were integrated with the shape file of the districts and their respective kebeles to produce the watershed's malaria risk map showing susceptibility level as high and moderate risk. Moreover, tables were also used to describe the map depictions.

The Fig 2 flow chart is used to show the general procedures applied in this study.

## Ethical consideration

The ethical issues related to research subjects have owed consideration in this study. The research protocol was reviewed and approved by Research Approval Committee of Wallagga University. The letter issued by the committee was used to request permissions from concerned bodies in the study area before the commencements of observatory data collection. The focus of informed consent was realized during data collection through discussions with the local community during the observations on the purpose of the research, duties, and responsibilities of the researcher. The letter was submitted to the participants and verbal consent was requested by reading the letter for those respondents cannot read the letter. Confidentiality and privacy of the local community regarding observable sociocultural entities have been realized that the observer was kept all issues confidential. All data used for the figures in this study are open and freely accessible. The biophysical data used in this study were prepared by the Ethiopian Central Statistical Agency, and other collaborative institutes. The data are freely accessible in universities and research institutes for research purposes and other academic exercises.

## Results

### Physical factors based malaria hot spot of the study area

Based on the value assigned, the amount and distribution of rainfall experienced in the watershed show spatial variations for malaria breeding suitability. Therefore, the ideal place for rainfall-based mosquito breeding is the area that lies within 1601-1723mm of rainfall which is assigned as the highest rainfall-based malaria risk area followed by rainfall 1504-1600mm. There is no place with low and very low capacity for malaria breeding. In general, as indicated on the map below (Fig 3A), 72.7% of the total watershed lays the highest rainfall-based malaria risk area; while the left 27.3% are within the moderately risk area (Table 3).

The area within the altitude of 1317–1500 masl was assigned as the highest elevation-based malaria hotspot followed by 1501–2000 and >2000 as high and moderate respectively. As altitude increases or decrease from this standard its suitability for malaria breeding decreases. Thus, these classifications were made to show areas with very high, high, and moderate altitude-based malaria risks in the watershed (Fig 3B). About 30.9%, 68.7%, and 0.4% of the watershed lie within very high, high, and moderate elevation-based malaria risk (Table 3).

Temperature is one of the determinant factors facilitating malaria breeding in the study area. According to criteria stated in different literature, a temperature between 21-24˚c is the most suitable for malaria survival followed by 19-20˚c which is assigned as high. However, according to the data on hand, the region lies totally within 19˚c (Fig 3C). Therefore, it is possible to conclude that the watershed has a high vulnerability to temperature-based malaria.

The slope of the study area was broadly categorized into five as 0–5, 5–8, 8–15, 15–30, and above 30 by which the values were assigned as very high, high, moderate, low, and very low respectively according to their ability to facilitate mosquitos breeding. Therefore, as indicated below (Fig 4A), 43.2% of the watershed lies within 0–5 slope and the area is characterized as a very high slope-based malaria risk area. Similarly, 31.6%, 22.7%, 2.4%, and 0.1% are believed

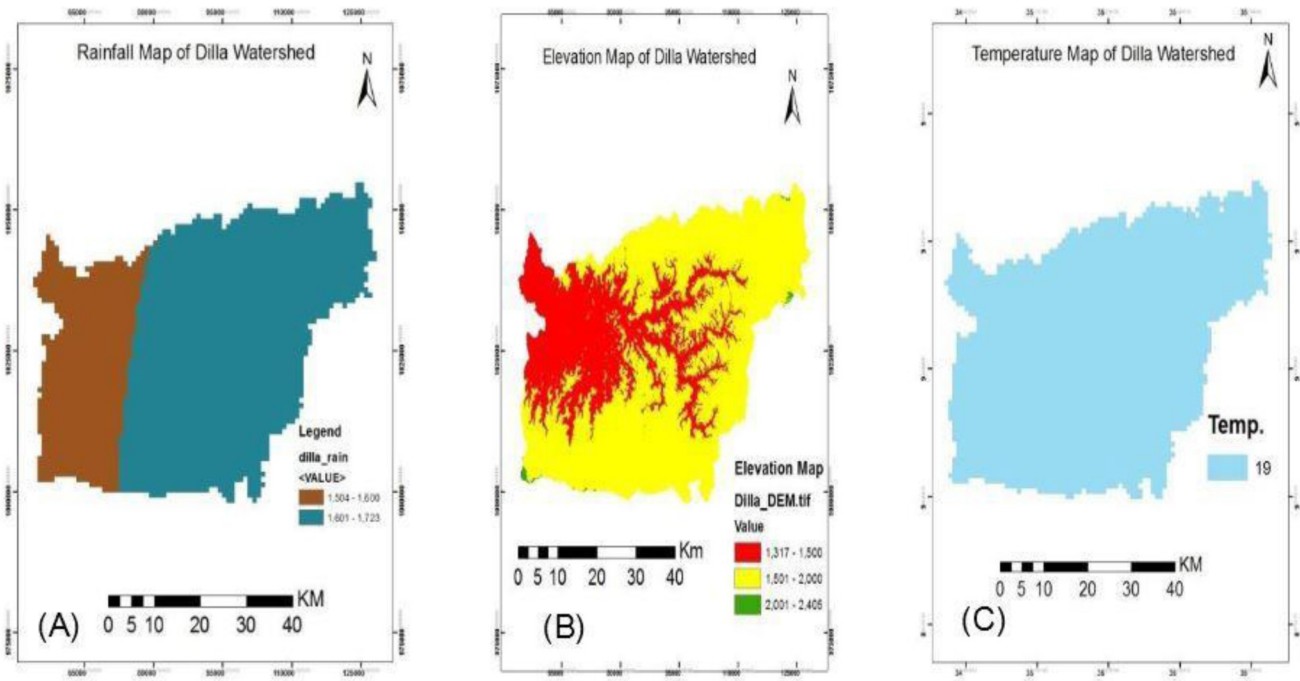

**Fig 3. Rainfall, Elevation, and Temperature Based Malaria Risk Area of Dilla Watershed (left to right). Source: Ethiopian Central Statistical Agency.**

**Table 3. Summary of the malaria risk level along the variables.**

| S.N | Variables | Risk Level | Area(Km$^2$) | Percentage |
|---|---|---|---|---|
| 1 | Rainfall | Very High | 2016 | 72.7 |
| | | High | 757 | 27.3 |
| | | Total | 2773 | 100 |
| 2 | Altitude | Very High | 856.9 | 30.9 |
| | | High | 1905 | 68.7 |
| | | Moderate | 11.1 | 0.4 |
| | | Total | 2773 | 100 |
| 3 | Temperature | High | 2773 | 100 |
| 4 | Slope | Very High | 1197.9 | 43.2 |
| | | High | 876.3 | 31.6 |
| | | Moderate | 629.5 | 22.7 |
| | | Low | 66.5 | 2.4 |
| | | Very low | 2.8 | 0.1 |
| | | Total | 2773 | 100 |
| 5 | Soil Drainage | Very poor | 58.2 | 2.1 |
| | | Poor | 38.8 | 1.4 |
| | | Moderate | 2676 | 96.5 |
| | | Total | 2773 | 100 |
| 6 | Proximity to Rivers | Very High | 403.8 | 24.4 |
| | | High | 364.1 | 22 |
| | | Moderate | 887.1 | 53.6 |
| | | Total | 1655 | 100 |

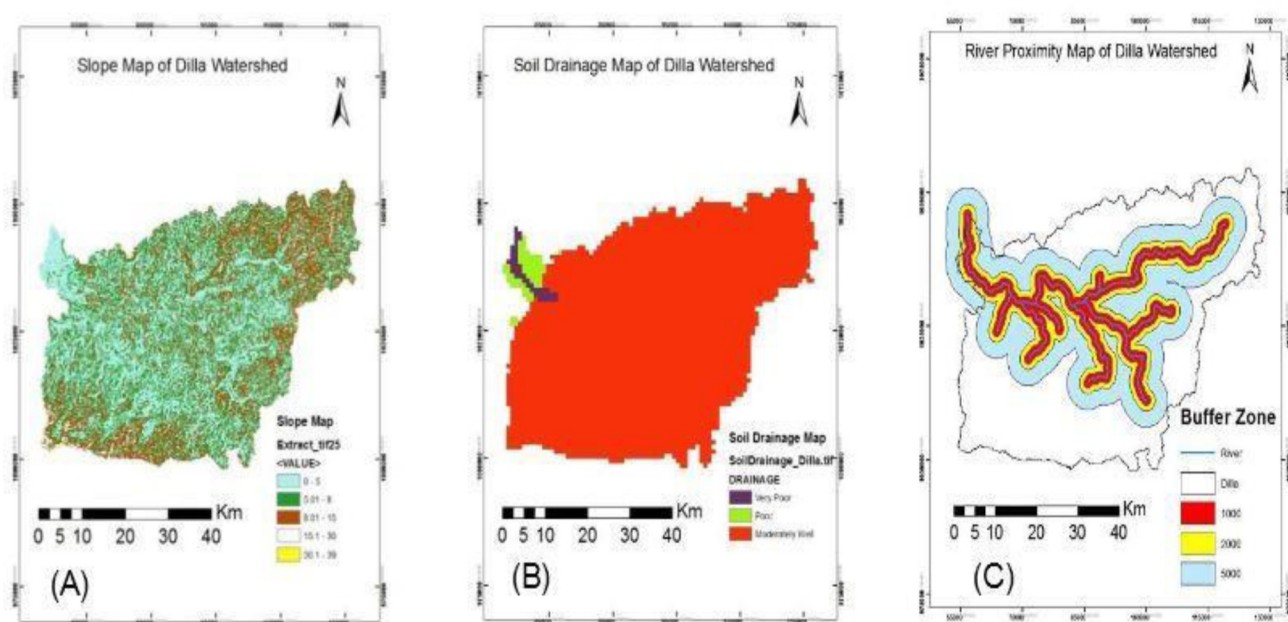

**Fig 4. Slope, soil drainage, and river proximity based malaria risk area of Dilla watershed.** Source: Ethiopian Central Statistical Agency.

to be high, moderate, low, and very low slope-based malaria risk areas. This indicated that the slope of the watershed is very susceptible for mosquito breeding.

Soil water holding capacity determines mosquitos' breeding. Therefore, soils with greater water retention capacity facilitate malaria prevalence in the study area. Accordingly, the highest percentage (96.5%) of the watershed is characterized by moderately well-drained soil types, only 2.1% and 1.4% are poor and very poor in soil drainage respectively (Table 3). Therefore, the soil drainage-based malaria risk was relatively low in the study area compared with other variables (Fig 4B).

The proximity to water-body has direct implication on the community's exposure to malaria hazards. The region closer to water body and swampy area has greater malaria breeding and flying zone risk. Thus, based on the criteria already identified, the study watershed has classified to closest region was assigned with 1kilometer buffering from the river, 1–2 kilometer and 2–5 kilometer have moderate and low exposure respectively (Fig 4C). Accordingly, only 59.5% of total areas of the watershed lie within the buffering of 5Km which are subjected to malaria risk of different levels. Thus, out of the total malaria vulnerable area (1655 Km2) the region most vulnerable due to its closest distance to the rivers (<1 Km) account 24.4% followed by medium distance (1–2 Km) accounts 22% and relatively far from the rivers (2–5 Km) which account 53.6% (Table 3).

## Malaria distributions and risk level of the watershed

The malaria risk map is the result of the overlaid maps of all variables according to their impact factors (assigned weight). The administrative map of the watershed was also integrated into the risk map to identify the level of districts and their respective kebeles risk to malaria hazards which helps the decision makers to identify areas requiring greater priority to the health facilities in the study area. As explained in the description of the study area, the watershed consists of 12 districts eight of which are in the west Wallaga zone and four in the Qellem Wallaga zone (Table 4).

The malaria risk map of the watershed indicated that all districts and kebeles have a different level of malaria risk due to variations in their physical settings affecting mosquito breeding. Accordingly, significant numbers of kebeles in Guliso, Aira, Boji Chokorsa, Boji Dirmaji, Jarso, Najo, Gawo Kebe, Dale Wabera, and others are high and moderately affected by malaria risk (Fig 5). Moreover, it is possible to prioritize the districts highly affected by malaria depending on the numbers of kebeles that lies within high and moderate risk level (Table 4).

In general, out of the total area of the watershed which accounts for 2773 km$^2$, 54.8% (1522km$^2$) was identified as a high and moderate malaria risk area. The details of districts and kebeles risk levels are described in Table 3 below.

## Discussions

The biophysical parameters identified in this study were found to have effects on the spatial variation in malaria prevalence, which corroborated prior studies done in different parts of Ethiopia and other regions of the world that came to similar results [21, 30]. The study found that the western and southwestern areas of the watershed had the highest malaria risk, whereas the risk decreased in the eastern part due to biophysical variations. Previous studies that used both historical clinical data of malaria incidences and geospatial models confirm a similar pattern and strong positive association between clinical admissions and biophysical-based hot spot situations [31, 32]. This indicates the role of spatial epidemiology in malaria prevention and management. Specifically, the study has also shown that the risk of malaria in the Dilla watershed declines with increasing altitude while altitude is the most determinant factor for

**Table 4. Malaria risk level of Dilla watershed in districts and kebele.**

| S.N | Districts | Risk Level | List of Kebele |
|---|---|---|---|
| 1 | Aira | High | Kure, LaloSuchi, WarababoSuchi, WarakuraSuchi, Norther part of WarawayuGerjo,WayuManni Sanki, HomiSuchi, Northern part of DagagaAira, Western part of Katta Abba Korma, TegiGallawo |
| | | Moderate | Central part of Kure and Northern Lalosuchi |
| 2 | Guliso | High | GawoGanka, Bedas Dilla, HawateSuchi,SekaJirbi,Maru, Sanki, ChaliyaKusaye, ChaliyaWaraDalle,JarsoBadeso, JarsoLalo, |
| | | Moderate | Some parts of Bedas Dilla, GawoGanka |
| 3 | BaboGambel | High | Ambalo Dilla, southern half of WarajiruBako |
| | | Moderate | MalkaEbicha |
| 4 | Jarso | High | Abono Dilla, GedoArgame,SouthernTuquSadan, south of GebaDafino,HidebuNyaha, BaboTiruben and Bedas Dilla |
| | | Moderate | Central part of GedoArgame and TuquSadan |
| 5 | Najo | High | Gunde Mikael, Didis Dilla, Gute Dilla, AmumaDilla,GamtaAmuma |
| | | Moderate | HomiGormitu, Lalistu Dilla, GidaKumbi |
| 6 | BojiChokorsa | High | Waligalte Dilla, TulluGuracha, western tip of Abo Makajarte, Western half of ChaliyaWara Ilu |
| | | Moderate | |
| 7 | BojiBirmaji | High | Eastern half of Bikiltu Dilla, AmomaBore,AgaloSilkan, west of AmumaAgelo |
| | | Moderate | North of DidibeTuli, east of AgaloSilkan, western half of Bikiltu Dilla |
| 8 | Yubdo | High | Some Kebeles of Yubdo woreda found at south eastern tip of the watershed is characterized by high altitude. Thus no Kebeles in the woreda has identified as high and moderate risk of malaria. |
| | | Moderate | |
| 9 | GawoQebe | High | North-east of Sichawo, North of Bata Dale and northern tip of BiftuChiro |
| | | Moderate | Northern part of BiftuChiro and Sichawo, eastern part of central Habro, centeral Bata Dale |
| 10 | Dale Wabera | High | Northern part of Haroji Hobo and AdanoWal-bata |
| | | Moderate | Scattered part of Haroji Hobo and AdanoWal-bata |
| 11 | Dale Sadi | High | Kebeles of Dale Sadi woreda found at southern tip of the watershed is characterized by high altitude. Thus no Kebeles in the woreda has identified as high and moderate risk of malaria. |
| | | Moderate | |
| 12 | LaloQile | High | Sarba Rajo is the only kebele found in this watershed with no high and moderate malaria risk level |
| | | Moderate | |

spatial variation in other biophysical factors. Similar studies also noted that as one goes up in altitude, the transmission potential falls off tremendously [5, 6, 11].

The significant difference between the result of spatial hotspot analysis and clinical data of malaria incidences caution us to consider vulnerability analysis to pinpoint the real and pressing factors for the effects on malaria on the local community [17, 29]. It has been suggested that this testimony is not the only mechanism to identify the spatial situation of malaria risks. Rather, the malaria risk maps ought to be produced by considering other socioeconomic and demographic factors are very relevant. Although such hotspot analysis are important, malaria vulnerability analysis become resonant issue than before. Second, future studies need to uses greater, sharper and finer spatial and temporal resolutions of biophysical risk factors. Thirdly, the physical factors such as climate and geomorphologic variability as a natural variable should be considered for their dynamism, that impacts on the conditions that affect mosquito proliferation. Moreover, this dynamism also includes human induced variables like land cover/ land use changes.

## Conclusions

From the study case, it can be inferred that there is a wide spectrum of possibilities over which geospatial models can be attributed to malaria control measures. Its application as an operational planning aid is an extension of geographical reconnaissance to promote better program management at both the local and national levels.

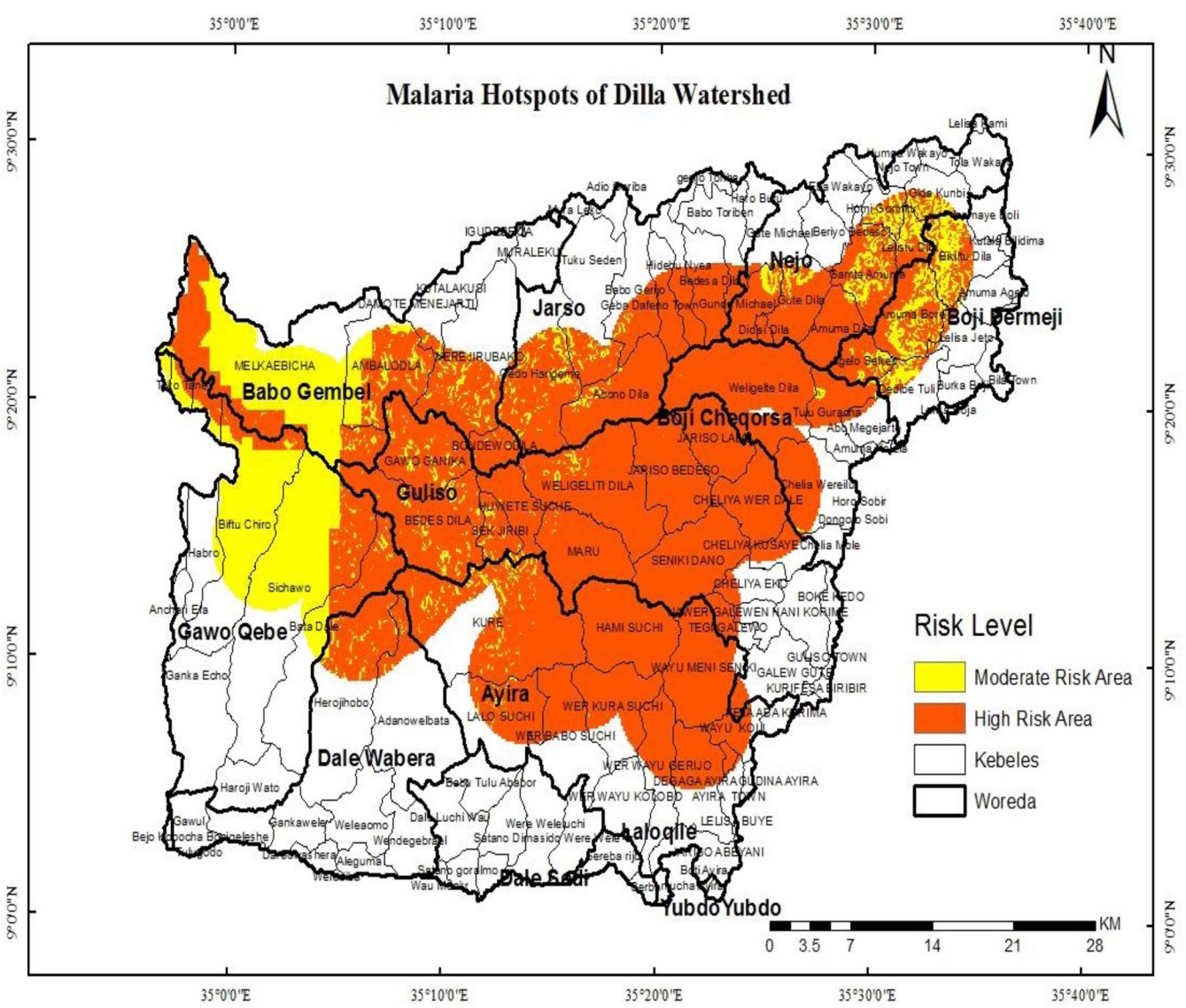

**Fig 5. Malaria risk map of Dilla watershed.** Source: Ethiopian Central Statistical Agency.

True to the objective of the study, malaria risk levels of the watershed were identified at the district and kebele levels. Physical factors used to generate a malaria risk map of the study area include temperature, rainfall, elevation, slope, soil drainage, and proximity to rivers. Maps were generated for all environmental factors and then reclassified based on their suitability for mosquito breeding and malaria incidence.

The final map of the malaria risk level of the watershed indicates that 54.9 percent of the total area of the watershed lies within high and moderate malaria risk levels. This is estimated to be 1522km$^2$ of land area. Out of the total areas, 82.2% lie within high malaria risk while the rest 17.8% is within moderate risk level.

At district level, significant areas of Aira, Guliso, Boji Birmaji, Boji Cokorsa, Najo and Jarso are found in high and moderate malaria risk level. More specifically, about 14 kebeles of Aira woreda, 12 kebeles of Guliso districts, 8 kebeles of Jarso districts, 8 kebeles of Najo districts, 4 kebeles of BojiChokorsa districts, 4 kebeles of Boji Birmaji districts and others are highly and moderately affecting by malaria prevalence.

The health office of West Wallagga and Qellem Wallagga zones are required to prioritize the areas of interventions based on the risk level at districts and kebele to provide malaria drugs, bed net distribution, and house spraying. This study was aimed only at malaria hotspot analysis. However, the map generated can be integrated with socioeconomic and other data to provide an excellent model for malaria disease management.

## Supporting information

**S1 Text. English language ground truthing (observation) data collection checklist.** (DOCX)

## Acknowledgments

This work was supported by Wallagga University, Ethiopia. I thank the University for the support to undertake this study. The biophysical data and the shape file of the study area were obtained from the Ethiopian Central Statistical Agency. I would like to appreciate the sources.

## Author Contributions

**Conceptualization:** Gemechu Y. Ofgeha.

**Data curation:** Gemechu Y. Ofgeha.

**Formal analysis:** Gemechu Y. Ofgeha.

**Investigation:** Gemechu Y. Ofgeha.

**Methodology:** Gemechu Y. Ofgeha.

**Project administration:** Gemechu Y. Ofgeha.

**Resources:** Gemechu Y. Ofgeha.

**Software:** Gemechu Y. Ofgeha.

**Supervision:** Gemechu Y. Ofgeha.

**Validation:** Gemechu Y. Ofgeha.

**Visualization:** Gemechu Y. Ofgeha.

**Writing – original draft:** Gemechu Y. Ofgeha.

**Writing – review & editing:** Gemechu Y. Ofgeha.

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
