## [Decision Letter · Decision Letter 0]

21 Sep 2022

PONE-D-22-21842Spatial Analysis of Malaria Hotspots in Dilla Sub-Watershed: Western EthiopiaPLOS ONE

Dear Dr. Ofgeha,

Thank you for submitting your manuscript to PLOS ONE. After careful consideration, we feel that it has merit but does not fully meet PLOS ONE’s publication criteria as it currently stands. Therefore, we invite you to submit a revised version of the manuscript that addresses the points raised during the review process.

 ======================Dear Authors, As you shall see that the reviewers have now commented on your manuscript. They are suggesting a major revision. Please go through the suggested revisions and revise your manuscript accordingly. Along with the revised manuscript files, please also submit the response to reviewers comments file also.

Best regards

Gowhar Meraj======================

We look forward to receiving your revised manuscript.

Kind regards,

Gowhar Meraj, Ph .D.

Academic Editor

PLOS ONE

Journal Requirements:

"This work was supported by Wollega University, Ethiopia."

4. We note that you have referenced [ie. Mestewat, S. (2014)] which has currently not yet been accepted for publication. Please remove this from your References and amend this to state in the body of your manuscript: (ie “Mestewat, S. [Unpublished]”) as detailed online in our guide for authors

5. We note that Figures 1 to 4 in your submission contain map images which may be copyrighted. All PLOS content is published under the Creative Commons Attribution License (CC BY 4.0), which means that the manuscript, images, and Supporting Information files will be freely available online, and any third party is permitted to access, download, copy, distribute, and use these materials in any way, even commercially, with proper attribution. For these reasons, we cannot publish previously copyrighted maps or satellite images created using proprietary data, such as Google software (Google Maps, Street View, and Earth). For more information, see our copyright guidelines: http://journals.plos.org/plosone/s/licenses-and-copyright.

a. You may seek permission from the original copyright holder of Figures 1 to 4 to publish the content specifically under the CC BY 4.0 license.  

Reviewers' comments:

Reviewer's Responses to Questions

**Comments to the Author**

1. Is the manuscript technically sound, and do the data support the conclusions?

Reviewer #1: Partly

Reviewer #2: Partly

2. Has the statistical analysis been performed appropriately and rigorously? 

Reviewer #1: N/A

Reviewer #2: Yes

3. Have the authors made all data underlying the findings in their manuscript fully available?

Reviewer #1: No

Reviewer #2: No

4. Is the manuscript presented in an intelligible fashion and written in standard English?

Reviewer #1: No

Reviewer #2: No

5. Review Comments to the Author

**Reviewer #1: **Thank you for the interesting work. A few suggestions are given to improve the readability of the manuscript.

1. The 'kebeles' need to be explained to the international audience, are they a sort of sub-regions/sub-blocks?

2. Expansion of the abbreviations like GDP(page 1), CSA(page7), etc should be given at the first mention.

3. Introduction could be more concise and it should substantiate the justification of the study. For example the study setting could be given as a separate section.

4. It seems there are many factual errors in the Conceptual Framework section.

-For example at one place it says 'the optional temperature for development of malaria parasite is between 25-30 degrees of Celsius(page 4, last para), but on page 5 second para, it says ' believing that temperature increasing more than 25 degree Celsius will gradually decrease the breading'.

-Again while describing elevation, the terms for the local vernacular, seems to rather confuse the readers.

-The units to be clearly spelled out in the first instance; what is meant by 1500masl?

-The classification of the slope classes and their names are not coinciding, kindly cross-check the order.

5. In figure 1, the component graphs need to be annotated, or proper titles to be given.

6. Method of data analysis need to be clearly stated, saying the utility function of a proprietary software is not enough.

7. The temperature is stated to be 19 degree Celsius through out the study area, but the number of centres reporting temperature was not given, was it a single centre?

8. The discussion session looks very weak, it needs to be expanded.

Finally I feel the manuscript should undergo a thorough language editing to bring in more clarity and remove the typing errors

**Reviewer #2: **The manuscript by Gemechu Y. Ofgeha provides important information of malaria hotspots under the moderate to high malaria transmission setting in Western Ethiopia using meteorological and altitude data as a main source for analysis. I have identified a number of issues that need to be addressed for further strengthening the manuscript as following:

ABSTRACT (Minor)

1. I suggest structure abstract which include: Introduction, Methods, Results and Conclusions

2. “other primary data” on line 5 should be listed out.

INTRODUCTION (Major)

Generally the introduction part needs major revision. To mention some area of revision

1. Same terms such as “Malaria hotspot”, “Biophysical” etc should be defined. Malaria hotspots occur at micro and macro geographical level, at what level the author wants to conduct malaria hotspot identification?.

2. Many information lack citation

3. The author used outdated data, for example “Malaria occurs in over 100 countries…World Health Organization estimates that 300-500 million cases of malaria occur worldwide each year result the death of over two million people.” This information also not cited. Another example “… an average of 5 million cases a year and 9.5 million cases per year between 2001 and 2005.” These data are almost 20 years old. The picture of malaria in recent years different in Ethiopia.

4. Most citation of references is not correct. For example, “According to Aynalem, Oromia region is one of the most populous and malaria-prone region in Ethiopia where three-quarters of the administrative districts (242 out of 261) and 3932 kebeles out of 6107 are considered malarious. Seventeen million people are at risk in the region with annual clinical cases numbering between 1.5 and 2 million (Aynalem, 2014).)”. There is no such information in “Aynalem, A. (2014). Ethiopian Demography and Health: A Brief Introduction. www.EthioDemographyAndHealth.Org”

5. The author used the term “incidence” in objective part of abstract and “prevalence” in objective of introduction part. These two words have different meaning the author should be consistent in using such words. He can use both words at the same time in his context.

6. In “Conceptual framework of the study”, categorization of variables such as “Rainfall”, “Slope”, “Proximity to Water Bodies”, and “Soil Moisture Holding Capacity” need appropriate citation.

7. The author should give equivalent definition of the five agro-ecological climatic zones in English/Amharic (official language of Ethiopia) that he mentioned as “Baddaa Dilallaa, Baddaa, Badda Daree, Gammojjii and Gammoojjii Ho’aa.”

8. There are many long, unclear sentences and incorrect grammar. Therefore, the language in the introduction part needs to be improved.

9. The is inappropriate words like “Malaria breeding”

 

METHODS AND MATERIALS (Major)

The methods and material parts are mostly well explained with some relevant uncertainties that have to be addressed. To mentioned some of them

1. “Figure 1: Map of the study Watershed” Need revision. E.g. Name of highlighted region (Oromia Region) in map of Ethiopia should be written in the map. Additionally, in the legend “Ethio_region, Oromia_Zone” should be removed.

2. The author claimed that he collected and used hospital and health center data to substantiate biophysical data, under “3.2 Research Design” and “3.3 Sources and Types of Data” parts. However, there are no such data in analysis and result parts of the study. If he used such data it needs detail description of the data, e.g. number of hospitals, health centers, and study period, total number of malaria cases and methods of analysis for such data to incorporate in malaria risk map etc.

3. The author also tried to show sources of data and analysis and outcome using a flow chart in “Figure 2”. In this part he mentioned that the key informants, health center data as data source. What data were collected from key informant and health centers? How this data incorporated in the analysis. The figure also misleads the readers.

4. Although the author claimed that the temperature is the same for all study area, I think there should be variation in temperature since as he mentioned altitude ranges between 1317 masl and 2405 masl in the study area.

5. How many years’ average data of temperature and rainfall used for this analysis?

6. At what level (kebele or district) the data available and used for analysis?

7. There was no ethical review and approval in the study. The author stated that he collected data from key informants.

RESULTS (Major)

The results are mostly well explained with some relevant uncertainties that have to be addressed. To mentioned some of them

1. In “Table 1” there is only high and very high level category. However, in the result part, the author used additional term “moderately high”. As I mentioned in method part, at what level this data mapped at kebele or districts?

2. The result in the maps in “Figure 2” not logical and convincing. Map in B (Altitude) affects both A (rainfall) and C (temperature). But, areas in South-West has high altitude ranges between 1500 to > 2000masl, but this area has similar rainfall with that of areas

< 1500masl. In C there is no variation in temp across study area. It is difficult to believe the truthfulness of data in this context.

3. I think “Map C” in Figure 3 is not correct. As author mentioned he made a buffer of <1km, 1-2km, and 2-5km. It seems the background map and buffers not proportional.

4. There is error in summation of numbers, e.g sum of areas Km2 under variable “Proximity to Rivers” is 1655 not 1650.

5. Under the sub title “4.2 Malaria Distributions and Risk level of the Watershed”, there is no data which shows malaria distribution. Therefore, this sub-title needs revision.

DISCUSSIONS (Major)

Generally the results poorly discussed, need major revision. To mentioned some points:

1. In the first sentence of discussion author tried to make association of biophysical parameters with spatial variation of malaria prevalence, and compared it with previous studies. However, he didn’t do such analysis and no such results.

2. The author clamed there about “significant difference between the result of spatial hotspot analysis and clinical data of malaria incidences” but he has no such data to discuss this issue.

CONCLUSION (minor)

The author recommends distribution of malaria drug, bed net distribution and house spraying for areas identified by his current study. I think without additional information like annual parasite incidence (API) of study area it is difficult to distribute resources only based on malaria risk map.

6. PLOS authors have the option to publish the peer review history of their article (what does this mean?). If published, this will include your full peer review and any attached files.

Reviewer #1: **Yes: **Dr Biju Soman

Reviewer #2: No

---

## [Author Response · Author response to Decision Letter 0]

3 Nov 2022

I appreciate you and the reviewers for your precious time in reviewing the paper and providing valuable comments. It was your valuable and insightful remarks that led to possible improvements in the current version. I have carefully considered the comments, incorporated the suggestions, and tried my best to address every one of them. I hope the manuscript after careful revisions meet your high standards. The detail is included in the "Responses to Reviewers"

---

## [Decision Letter · Decision Letter 1]

17 Jan 2023

PONE-D-22-21842R1Spatial Analysis of Malaria Hotspots in Dilla Sub-Watershed: Western EthiopiaPLOS ONE

Dear Dr. Ofgeha,

Thank you for submitting your manuscript to PLOS ONE. After careful consideration, we feel that it has merit but does not fully meet PLOS ONE’s publication criteria as it currently stands. Therefore, we invite you to submit a revised version of the manuscript that addresses the points raised during the review process.

We look forward to receiving your revised manuscript.

Kind regards,

Bijeesh Kozhikkodan Veettil

Academic Editor

PLOS ONE

Journal Requirements:

Reviewers' comments:

Reviewer's Responses to Questions

**Comments to the Author**

1. If the authors have adequately addressed your comments raised in a previous round of review and you feel that this manuscript is now acceptable for publication, you may indicate that here to bypass the “Comments to the Author” section, enter your conflict of interest statement in the “Confidential to Editor” section, and submit your "Accept" recommendation.

Reviewer #1: (No Response)

2. Is the manuscript technically sound, and do the data support the conclusions?

Reviewer #1: Partly

3. Has the statistical analysis been performed appropriately and rigorously? 

Reviewer #1: I Don't Know

4. Have the authors made all data underlying the findings in their manuscript fully available?

Reviewer #1: No

5. Is the manuscript presented in an intelligible fashion and written in standard English?

Reviewer #1: Yes

6. Review Comments to the Author

Reviewer #1: Thank you, the manuscript has substantially improved. Still the language does not have professional vibe. It would be better to get your manuscript, vetted by a language editor. the introduction session could be trimmed

7. PLOS authors have the option to publish the peer review history of their article (what does this mean?). If published, this will include your full peer review and any attached files.

Reviewer #1: No

---

## [Author Response · Author response to Decision Letter 1]

8 Feb 2023

Dear Editor-in-chief,

I appreciate you and the reviewer for your precious time in reviewing the paper and providing valuable comments. It was your valuable and insightful remarks that led to possible improvements in the current version. I have carefully considered the comments, incorporated the suggestions, and tried my best to address every one of them. I hope the manuscript after careful revisions meet your high standards. 

This response letter consist the responses on the comments provided by the editor and reviewers. I provide the point-by-point responses for the detailed comments and suggestions. I look forward to hearing from you in due time regarding the resubmission, and to respond to any further questions and comments you may have.

(Editor's Comments) Journal Requirements: Please review your reference list to ensure that it is complete and correct. If you have cited papers that have been retracted, please include the rationale for doing so in the manuscript text, or remove these references and replace them with relevant current references. Any changes to the reference list should be mentioned in the rebuttal letter that accompanies your revised manuscript. If you need to cite a retracted article, indicate the article’s retracted status in the References list and also include a citation and full reference for the retraction notice. 

(Response): Thank you very much. I have exhaustively reviewed the reference list that it is complete and correct; and no retracted citation was used in this manuscript. Because of some revisions mainly to trim the introduction part, new citations are used, and referenced. 

• Devi P, B M, S B. Use of Remote Sensing and GIS for Monitoring the Environmental Factors Associated With Vector-Borne Disease (Malaria). Proc Third Int Conference Environment

• Gebreslasie MT. A review of spatial technologies with applications for malaria transmission modelling and control in Africa. Geospat Health. 2015;10(2):239–47.

• Larocca A, Moro Visconti R, Marconi M. Malaria diagnosis and mapping with m-Health and geographic information systems (GIS): evidence from Uganda. Malar J. 2016; 15(1):1–12.

(Reviewer #1 Comment) “The manuscript has substantially improved. Still the language does not have professional vibe. It would be better to get your manuscript, vetted by a language editor. The introduction session could be trimmed”

(Response): Thank you for your positive response on the revision. Again in this revision, I have carefully considered the suggestions to improve the language problems. Now, I have used the English language profession in the host University for this ‘Revision’. True to the critical comments raised by the respected reviewer, I have found several grammatical errors. However, now the manuscript is carefully revised. The introduction part is exhaustively revised and trimmed.

---

## [Editor Report · Decision Letter 2]

20 Feb 2023

Spatial Analysis of Malaria Hotspots in Dilla Sub-Watershed: Western Ethiopia

PONE-D-22-21842R2

Dear Dr. Ofgeha,

We’re pleased to inform you that your manuscript has been judged scientifically suitable for publication and will be formally accepted for publication once it meets all outstanding technical requirements.

Kind regards,

Bijeesh Kozhikkodan Veettil

Academic Editor

PLOS ONE
---

## [Editor Report · Acceptance letter]

27 Feb 2023

PONE-D-22-21842R2 

Spatial Analysis of Malaria Hotspots in Dilla Sub-Watershed: Western Ethiopia 

Dear Dr. Ofgeha:

I'm pleased to inform you that your manuscript has been deemed suitable for publication in PLOS ONE. Congratulations! Your manuscript is now with our production department. 

Kind regards, 

on behalf of

Dr. Bijeesh Kozhikkodan Veettil 

Academic Editor

PLOS ONE